# Bi-MOF-Derived Carbon Wrapped Bi Nanoparticles Assembly on Flexible Graphene Paper Electrode for Electrochemical Sensing of Multiple Heavy Metal Ions

**DOI:** 10.3390/nano13142069

**Published:** 2023-07-14

**Authors:** Min Hu, Hu He, Fei Xiao, Chen Liu

**Affiliations:** 1Key Laboratory of Material Chemistry for Energy Conversion and Storage, Ministry of Education, School of Chemistry and Chemical Engineering, Huazhong University of Science & Technology, Wuhan 430074, China; m202170297@hust.edu.cn; 2Technology Inspection Center of Sheng Li Oil Filed, Dongying 257000, China; hehu.slyt@sinopec.com; 3Research Institution of Huazhong University of Science and Technology in Shenzhen, Shenzhen 518052, China

**Keywords:** Bi nanoparticles, nitrogen-doped carbon, flexible and freestanding graphene paper electrode, electrochemical catalysis and sensing, heavy metal ions detection

## Abstract

The development of nanohybrid with high electrocatalytic activity is of great significance for electrochemical sensing applications. In this work, we develop a novel and facile method to prepare a high-performance flexible nanohybrid paper electrode, based on nitrogen-doped carbon (NC) wrapped Bi nanoparticles (Bi-NPs) assembly derived from Bi-MOF, which are decorated on a flexible and freestanding graphene paper (GP) electrode. The as-obtained Bi-NPs encapsulated by an NC layer are uniform, and the active sites are increased by introducing a nitrogen source while preparing Bi-MOF. Owing to the synergistic effect between the high conductivity of GP electrode and the highly efficient electrocatalytic activity of Bi-NPs, the NC wrapped Bi-NPs (Bi-NPs@NC) modified GP (Bi-NPs@NC/GP) electrode possesses high electrochemically active area, rapid electron-transfer capability, and good electrochemical stability. To demonstrate its outstanding functionality, the Bi-NPs@NC/GP electrode has been integrated into a handheld electrochemical sensor for detecting heavy metal ions. The result shows that Zn^2+^, Cd^2+^, and Pb^2+^ can be detected with extremely low detection limits, wide linear range, high sensitivity, as well as good selectivity. Furthermore, it demonstrates outstanding electrochemical sensing performance in the simultaneous detection of Zn^2+^, Cd^2+^, and Pb^2+^. Finally, the proposed electrochemical sensor has achieved excellent repeatability, reproducibility, stability, and reliability in measuring real water samples, which will have great potential in advanced applications in environmental systems.

## 1. Introduction

Heavy metal pollution is a hazardous consequence of industrialization [1,2,3]. In contrast to organic pollutants, heavy metal elements are non-biodegradable, and once released into the environment through industrial wastewater and waste, they can accumulate along the food chain, leading to chemical contamination of food and posing a threat to public health [4]. Therefore, it is essential to reduce the discharge of heavy metal ions (HMIs) into the environment [5,6]. To achieve this, it is necessary to develop water monitoring techniques with high precision and ultra-sensitivity to detect HMIs [7,8]. It is well known that HMIs in natural water typically appear in a complex form [9,10], i.e., multiple heavy metal ions coexist to form intermetallic compounds that compete for adsorption at the active site, which leads to a more limited simultaneous detection. Therefore, it is necessary to develop reliable, ultra-sensitive, and real-time detection methods for the simultaneous detection of multiple HMIs for practical application. The methods currently used for HMIs detection are relatively mature, such as inductively coupled plasma–mass spectrometry [11], atomic absorption spectrometry [12], X-ray fluorescence spectrometry [13], and atomic emission spectrometry [14]. However, these methods are costly, require specialized personnel, and involve complex sample pre-processing, making them unsuitable for real-time in situ monitoring [6]. Due to the inherent advantages of low cost, fast response, easy-to-use and miniaturization, the electrochemical sensors can fit well to the simultaneous analysis of multiple HMIs [15,16]. As the core component of electrochemical sensors, the electrode materials play a decisive role in the performances of the sensing devices. Therefore, recent studies have focused on the design of advanced electrode materials for HMIs sensing.

Owing to the high electrocatalytic activity, excellent electrical conductivity and large specific surface area arising from their structure and size, metal and metal oxide nanoparticles have been leveraged in past decades for electrochemical detection of various hazardous materials including HMIs [17]. Amongst them, mercury (Hg) has been commonly used as the preferred material for electrodes in HMIs detection due to its ability to form an amalgam with one or several other metals [18]. Alternately, Bi has emerged as a promising alternative to Hg as it has low toxicity [19]. However, Bi nanoparticles (Bi-NPs) are usually prone to agglomeration or clustering, thereby compromising the sensitivity and stability of electrochemical sensors. To mitigate this problem, an effective strategy is to anchor the metal nanoparticles onto a suitable substrate, which could help in preventing the agglomeration and improving catalytic activity and stability. As a result, carbon materials are generally considered as substrates for loading metal nanomaterials due to their good conductivity and high stability [20,21]. Specifically, carbon nanomaterials synthesized by carbonization of metal–organic frameworks (MOFs) can inherit the layer nanostructure of their MOF precursors, resulting in highly conductive network, a large surface area, and good chemical and structural stability. Meanwhile, metal nanoparticles with good dispersion and uniformity can also be obtained during the process. The resultant metal–carbon nanohybrid materials derived from MOFs have tremendous potential for application in the field of electrochemical catalysis [22].

In this work, we developed a new type of Bi-NPs wrapped by nitrogen-doped carbon (NC) derived from Bi-MOF (i.e., bismuth benzenedicarboxylate, Bi-BDC), which is loaded onto a flexible and freestanding graphene paper (GP) electrode. The GP was synthesized by a scalable printing method according to our previous work [23], which possesses high electric conductivity, good stability, and adjustable electrode size. In addition, Bi-NPs wrapped by NC (Bi-NPs@NC) derived from Bi-BDC offer several advantages. This structure provides a high loading density of Bi-NPs to ensure their dispersion on the NC layer, effectively preventing the agglomeration effect commonly observed with Bi-NPs. Moreover, heteroatom-doped carbon can lead to the abundance of non-bonding pairs near the surface, which contributes to the enhanced ion adsorption, and thus enhances the catalytic activity of Bi-NPs in Bi-NPs@NC/GP for specific catalytic reactions [24]. To demonstrate its excellent functionality, we constructed a handheld electrochemical sensor based on Bi-NPs@NC/GP electrode for the detection of HMIs. The sensor enables the simultaneous detection of multiple HMIs with low detection limits (i.e., 10 ppb for Zn^2+^, 0.5 ppb for Cd^2+^, and 0.1 ppb for Pb^2+^, S/N = 3), the linear range is from 10 to 1200 ppb for Zn^2+^, from 0.5 to 1200 ppb for Cd^2+^, and as wide as 10 mmol L^−1^ for Pb^2+^ when detected individually (in the absence of other heavy metal ions), and the detection limit can reach 1 pmol L^−1^ under the optimal conditions. This strategy can effectively improve the electrocatalytic activity and sensitivity of nanomaterial-based electrodes, and the preparation process is simple and can be produced by the large-scale and low-cost method. Moreover, this portable sensor system can not only provide the feasibility for accurate and sensitive detection, but also enable the real-time on-site measurement in various environments.

## 2. Material and Methods

### 2.1. Materials

Bismuth nitrate pentahydrate (Bi(NO_3_)_3_·5H_2_O), sodium borohydride (NaBH_4_), and N,N-dimethylformamide (DMF) were purchased from Sinopharm Chemical Reagent Co., Ltd. (Shanghai, China). Terephthalic acid (H_2_BDC) and imidazole were purchased from Aladdin Reagent. The stock solutions of HMIs were prepared by dissolving CdCl_2_ (Tianjin Bo Di Chemical Industry Co., Ltd., Tianjin, China), PbCl_2_ (Tianjin Bo Di Chemical Industry Co., Ltd., Tianjin, China), and ZnCl_2_ (Adamas Reagent Co., Ltd., Shanghai, China) in 1% (*v*/*v*) nitric acid. The aqueous solutions were prepared with ultrapure water (≥18 MΩ cm^−1^) from a Millipore system, and a graphene solution of 5 mg mL^−1^ was prepared using a modified Hummer’s method (graphite oxide powders, 99.95%, Aladdin Co., Dubai, United Arab Emirates) [25].

### 2.2. Preparation of Nanocomposite Electrodes

The graphene oxide (GO) solution was synthesized using a modified Hummer’s method at a concentration of 5 mg mL^−1^. Then, GO paper (GOP) was prepared by the printing method according to our previous work [23]. The Bi-BDC was synthesized according to a previously reported work [26]. In the process, 0.6 mmol of Bi(NO_3_)_3_·5H_2_O, 3.0 mmol of H_2_BDC, and 2.4 mmol of imidazole were added to 10.0 mL of DMF and stirred for 30 min. The resulting mixture was then transferred to a stainless steel autoclave with a Teflon liner and heated at 120 °C for a duration of 24 h. The precipitate was collected and washed repeatedly with DMF and ethanol. Subsequently, Bi-BDC was dissolved in 20 mL of DMF, and GOP was soaked in the Bi-BDC solution for 24 h. Thereafter, the Bi-BDC modified GOP (Bi-BDC/GOP) was removed, washed several times with deionized water, and dried at 60 °C. Then, the as-synthesized Bi-BDC/GOP was placed in a tube furnace and subjected to a temperature heating process that was increased from room temperature to 600 °C at a rate of 10 °C min^−1^ (denominated as Bi-NPs@NC/GP-600 °C). Bi-NPs@NC/GP-500 °C, Bi-NPs@NC/GP-700 °C, Bi-NPs@NC/GP-800 °C, Bi-NPs@NC/GP-900 °C, and Bi-NPs@NC/GP-1000 °C were also prepared under the same procedures. The sample was held at different temperatures for 2 h, with protection provided by N_2_. Finally, the organic ligands were transformed from the MOF structure into NC materials, while the uniformly dispersed metal ions located at the nodes were reduced to Bi-NPs. Meanwhile, the GOP underwent complete reduction at high temperatures, leading to the final formation of Bi-NPs@NC/GP. For comparison, Bi-NPs/GP electrode was also synthesized. Typically, 0.486 g of Bi(NO_3_)_3_·5H_2_O was weighed and dissolved in 50 mL H_2_O, the mixture was stirred for 15 min, and 0.378 g of NaBH_4_ was then added and stirred for another 15 min. The resulting solution was poured into a reaction vessel, and GOP was immersed in the solution for a 24 h reaction at 160 °C. After completion of the reaction, the product was washed with deionized water at least three times and dried at 60 °C.

### 2.3. Fabrication of Hand-Held Integrated Sensing Equipment

The Bi-NPs@NC/GP material is clamped into an electrode sleeve as the working electrode, with Ag/AgCl as the reference electrode and a Pt sheet as the counter electrode. These electrodes are assembled and enclosed within a glass sleeve with a sealed hole at the bottom. This design makes it possible to directly measure the three-electrode system within the water bodies. To prevent damage by contaminants presented in the water samples, a sealed hole spacer is added to fix the anti-fouling film (Appendix A). A micro-electrochemical workstation is used to collect the electrical signal outputs from the sensor, which is connected to Bluetooth technology, wireless networks, and mobile terminal devices.

### 2.4. Electrochemical Measurement of HMIs by Bi-NPs@NC/GP

The three-electrode sensing system is immersed in a 0.1 mol L^−1^ (0.1 M) acetate buffer solution (ABS) at pH 5. After optimizing the detection conditions, metal ions from the analyte samples were pre-deposited at an electrodeposition potential of −1.2 V for 180 s. Parameters are set for square wave stripping voltammetry (SWASV) measurements as follows: An initial voltage of −1.4 V, a final voltage of −0.3 V, a frequency of 15 Hz, an amplitude of 25 mV, and a potential increment of 4 mV. To reuse the electrode, a dissolution step is conducted by applying a potential of 0.3 V for 80 s to remove excess metal from the electrode surface. Every experiment is repeated at least three times, and the data are presented as mean ± standard deviation (SD).

### 2.5. Treatment of Real Samples

Natural water samples were collected from tap water, Wuhan East Lake, and Yujia Lake. All samples were pretreated by centrifugation at 10,000 rpm for 20 min, then filtered through a 0.22 μm membrane to remove insoluble impurities, and diluted 10 times with 0.1 M ABS (pH 5).

### 2.6. Acquisition of ICP-MS Data for Real Samples

The ICP-MS data were collected by preparing a certain concentration of ion standard solution, and calibrating it by ICP-MS to obtain a standard curve. Then, we filtered and diluted the practical samples for ICP-MS measurement, and obtained their concentration values from intensities.

## 3. Results

### 3.1. Preparation and Characterization of Bi-NPs@NC/GP

In the process of preparing Bi-NPs@NC/GP electrode, Bi-BDC was first obtained by solvent-thermal treatment of H_2_BDC, HIM, and Bi(NO_3_)_3_. Subsequently, it was successfully loaded onto GOP via the room-temperature immersion method. During this process, Bi-BDC nanomaterials were anchored onto the surface of GOP. After calcination at high temperature, Bi-BDC was transferred into Bi-NPs@NC and GOP was thermally reduced to GP with high electronic conductivity and stability, as shown in Figure 1.

The microstructure and morphology of the obtained materials are characterized using scanning electron microscopy (SEM) and transmission electron microscopy (TEM). Figure 2A shows that the GO nanosheets exhibit typical wrinkle-like morphology, which can be further assembled into macroscopic freestanding GOP (Figure 2B), with well-layered structure throughout its cross section (Figure 2C). Figure 2D shows that Bi-BDC particles exhibit regular rectangular shapes [27]. By an ambient-temperature immersion approach, Bi-BDC particles are successfully loaded onto GOP to form multi-layer GO-MOF hetero-aggregates with high geometric stability (Figure 2E). Upon high-temperature calcination, the organic ligands of Bi-BDC structure are converted into NC materials, while the uniformly dispersed metal ions at nodes are reduced into Bi-NPs to form unique Bi-NPs@NC assembly (Figure 2F). This structure will effectively prevent the agglomeration of Bi-NPs, thereby greatly improving the electron transfer efficiency of metal nanocatalyst. Furthermore, after annealing, Bi-NPs ranging in size from 10 to 50 nm are uniformly distributed on NC layer (Figure 2G), leading to an increased contact area with the target HMIs. This is beneficial for improving the electrical sensing performance. Figure 2H,I show that Bi-NPs have been anchored in the NC layer, as indicated by a lattice plane spacing of 0.327 nm corresponding to the (012) plane of Bi [28], and a lattice plane spacing of 0.34 nm corresponding to the (002) plane of C [29]. The surrounding hexagonal rings in the carbon domain derived from graphene tightly surround the Bi-NPs [30]. This stable chemical bond will enhance the stability of the electrode during testing and the affinity with the target ions. In addition, the energy dispersive spectroscopy (EDS) mapping image shows the spatial distribution of Bi and C elements in Bi-NPs@NC material, which confirmed the formation of Bi-NPs on carbon nanosheets (Figure 2J).

Figure 3A displays X-ray diffraction (XRD) patterns of different samples. For Bi-BDC/GOP, the characteristic peak of graphite appeared at ~10° and ~26° [31], and that of Bi-BDC is consistent with the simulated peaks. Both Bi-NPs@NC/GP and Bi-NPs@NC exhibit a similar crystal structure that was consistent with the crystal structure of Bi (JCPDS No. 44-1246). The diffraction peaks of Bi (012) and Bi (104) were located at ~28° and ~38°, respectively [32,33]. It should be noted that in all XRD results, the 2θ peaks nearby 25°–35° are relatively wide. This could be attributed to the slight oxidation of the surface of Bi-NPs. The mention of “oxidation” here does not refer to the presence of bismuth oxide. If bismuth oxide was present, it would give rise to new peaks rather than broadening the Bi peak. Additionally, the limited number of coherently scattering lattice planes can cause a peak widening, which is a common phenomenon observed in nanocrystalline materials, such as the Bi particles under investigation. Furthermore, there are no clear carbon diffraction peaks detected, indicating that the obtained carbon is amorphous. As depicted in the X-ray photoelectron spectroscopy (XPS) survey spectrum (Figure 3B), the Bi-NPs@NC/GP contains Bi, C, O, and N elements, which exactly match with the TEM elemental mapping result. The high-resolution C 1s spectrum of the Bi-NPs@NC/GP shown in Figure 3C exhibits three decoupled peaks of C–C (284.8 eV), C–N (285.4 eV), C–O (286.5 eV), and C=O (289 eV), respectively [34]. The O 1s spectra can be deconvoluted into Bi-O (530.4 eV), –O–H bonds (532.3 eV), and C–O (533.8 eV) [35], as shown in Figure 3D. Moreover, in this work, Bi-MOF was prepared by adding imidazole, and the N of imidazole would coordinate with Bi during the self-assembly process of Bi-MOF. After calcination at high temperature, Bi-MOF (i.e., Bi-BDC) was transferred into Bi-NPs wrapped by N-doped carbon. As shown in the N 1s scan spectrum in Figure 3E, the N 1s spectra can be compartmentalized into three kinds, including pyridinic N (398.6 eV), graphitic N (400.4 eV), and oxidized N (402.2 eV) [36], indicating the successful doping of N element. The atomic concentration of N in Bi-NPs@NC/GP is 1%, which is regarded as the efficient active site for electrocatalysis. Evidently, heteroatom dopants (i.e., N) play a crucial role in effectively modulating the electrical properties and surface physicochemical characteristics of carbon materials, which results in enhanced activity or the introduction of new chemical functions [37]. Notably, in the high-resolution Bi 4f spectrum, two peaks are observed at 159.5 and 164.8 eV (Figure 3F), which shifted toward higher values than the bulk Bi [38]. This suggests that thermodynamically-stable covalent bonds between Bi and NC are formed, creating a stable interface between the Bi-NPs and NC, rather than a simple mechanical contact between them (similar to that of the Bi/GP sample). These geometric and structural characteristics can excellently promote electron transfer and maximize the exploitation of Bi while forming alloy sensing with HMIs.

### 3.2. Electrochemical Characterization of Bi-NPs@NC/GP

The electrochemical properties of different electrodes are first investigated by cycle voltametric (CV) measurements using 5 mM [Fe(CN)_6_]^3−/4−^ as the probe. As demonstrated in Figure 4A and Appendix A, a pair of symmetrical redox peaks appeared on the CV curves of Bi-NPs@NC/GP, Bi-NPs@GP, and Bi-BDC/GOP, which result from the reversible single electron redox reaction of [Fe(CN)_6_]^3−/4−^. The electrochemically active surface areas (ECSAs) of different modified electrodes are calculated by the Randles-Sevcik equation, which are 0.028, 0.036, and 0.12 cm^2^ for Bi-BDC/GOP, Bi-NPs/GP, and Bi-NPs@NC/GP, respectively. In addition, the Bi-NPs@NC/GP demonstrates a peak-to-peak potential separation (ΔE_p_) of 69 mV, whereas that of Bi-BDC/GOP and Bi-NPs/GP are 156 mV and 141 mV, respectively, suggesting that Bi-NPs@NC/GP would greatly accelerate the heterogeneous electron transfer between the electrode and the [Fe(CN)_6_]^3−/4−^ redox species. Figure 4B shows the characteristic Nyquist plots of Bi-BDC/GOP, Bi-NPs/GP, and Bi-NPs@NC/GP, fitted with the Randles equivalence circuit inset, where R_s_ is the resistance of solution (the intersection of the curve at real part Z_0_ in the high frequencies range), R_ct_ is the Faradic charge transfer resistance (the semicircle at high frequencies), C_dl_ is the double layer capacitance, and W is Warburg constant (the slope of the curves at a low frequency) [37]. The R_ct_ values are calculated to be 1315 Ω, 861.9 Ω, and 285.7 Ω for Bi-BDC/GOP, Bi-NPs/GP, and Bi-NPs@NC/GP, respectively, indicating that Bi-NPs@NC can effectively promote the electron transfer. To further demonstrate the electrochemical activity of Bi-NPs@NC/GP in detection of HMIs, its electrochemical performance has been characterized using SWASV in 0.1 M ABS (pH 5) containing 400 ppb Cd^2+^. Figure 4C shows a strong stripping peak that appeared at −0.75 V on Bi-NPs@NC/GP for Cd^2+^ [24], and peak current densities are significantly higher in comparison with that of Bi-NPs/GP. Benefitting from the uniform loading of Bi-NPs on carbon nanomaterials, as well as the large specific surface area and heteroatom doping, Bi-NPs@NC/GP leads to abundant electrochemically active sites, efficient and stable charge channels and faster electron transfer rate, and high electrocatalytic stability, which effectively improve the sensing performances of Bi-NPs@NC/GP.

### 3.3. Optimization of Experimental Conditions

Since the calcination temperature for the synthesis of electrode materials may affect the size and structure of Bi-NPs, we investigated the electrochemical sensing performances of different electrodes (i.e., Bi-NPs@NC/GP-500 °C, Bi-NPs@NC/GP-600 °C, Bi-NPs@NC/GP-700 °C, Bi-NPs@NC/GP-800 °C, Bi-NPs@NC/GP-900 °C, and Bi-NPs@NC/GP-1000 °C). As shown in Figure 5A, the maximum oxidative dissolution peak current of HMIs is observed for Bi-NPs@NC/GP-600 °C. The possible reason for this phenomenon may be that at calcination temperatures below 600 °C, the size of Bi-NPs is relatively large and their distribution is not uniform. However, when the temperature exceeds 600 °C, the structure of the nanoparticles may collapse. In addition, the effect of the pH value of the buffer solution has been evaluated (Figure 5B), and the results suggest that the optimal pH ranging from 3.0 to 7.0 is 5.0. The possible reason for the decrease in peak currents at relatively lower pH values (i.e., pH 3.0 to 5.0) is due to the protonation of hydrophilic groups, which weaken the absorption of HMIs to the electrode materials [39]. At relatively high pH values (i.e., pH 5 to 7), the decrease in current signal is attributed to the hydrolysis of HMIs, which hinder the accumulation of HMIs at the electrode surface [40,41]. Figure 5C shows that the peak current densities of Zn^2+^, Cd^2+^, and Pb^2+^ increase as the deposition potential shifts from −0.6 to −1.2 V. This is due to the fact that a negative shift in the deposition potential encourages the reduction in HMIs, leading to a significant increase in peak current density. However, the peak current densities decrease when the deposition potential shifts from −1.2 to −1.4 V, which is due to the hydrogen evolution reaction that produces hydrogen bubbles and causes damage to the metal deposited on the electrode surface [42]. Furthermore, the electrodeposition time was optimized. The result demonstrates that the intensity of the current signal increases with an increase in deposition time and reaches an approximate maximum after around 180 s (s), remaining essentially stable thereafter (Figure 5D). Therefore, the optimal experiment time of 180 s is used considering reliability and the need for a rapid measurement.

### 3.4. Analytical Performances of the Integrated Electrochemical Sensor

The analytical performances of Bi-NPs@NC/GP for the individual detection of Zn^2+^, Cd^2+^, and Pb^2+^ were evaluated using SWASV under optimal conditions. The results show that the current responses of each HMI display a linear increase as the concentration of the target analyte increased (Figure 6A–C and Appendix A). The linear ranges are 10–1200 ppb for Zn^2+^ and 0.5–1200 ppb for Cd^2+^. Remarkably, the linear range for the detection of Pb^2+^ is exceptionally broad, covering a range from 0.5 ppb to 2.1 g L^−1^, equivalently up to 10 mmol L^−1^. Figure 6D–F show the linear equations for the Bi-NPs@NC/GP based sensor toward Zn^2+^, Cd^2+^, and Pb^2+^, and the results are summarized in Table 1. The outstanding analytical sensitivity of the Bi-NPs@NC/GP sensor is further evidenced by its low limit of detection (LOD), which are found to be 5 ppb for Zn^2+^, 0.05 ppb for Cd^2+^, and 0.02 ppb for Pb^2+^ as determined through a signal-to-noise ratio of 3 (S/N = 3). Additionally, the Bi-NPs@NC/GP has the ability to detect Pb^2+^ at a remarkably low level of 1 pmol L^−1^ with an extended electrodeposition time. It is noteworthy that the LOD achieved by the sensor is found to be lower than the WHO recommended guideline values for drinking water (Zn^2+^: 1.0 mg L^−1^, Cd^2+^: 0.003 mg L^−1^, Pb^2+^: 0.01 mg L^−1^), underscoring the practical potential of the integrated sensor for HMIs detection.

The detection of target HMIs with high sensitivity and accuracy remains a significant challenge in analyzing real samples, where various HMIs are often present simultaneously [41]. This is particularly challenging due to the mutual interference of these ions, which can significantly affect the accuracy and reliability of HMIs detection methods. In this work, the simultaneous detection of Zn^2+^, Cd^2+^, and Pb^2+^ was performed under optimized conditions by increasing the concentrations of different HMIs synchronously within the potential range of −1.4 to −0.3 V. Figure 7A shows that the dissolution peak currents of Zn^2+^, Cd^2+^, and Pb^2+^ increase as the concentration of HMIs increases within the range of 1–1400 ppb for Cd^2+^ and Pb^2+^, and 20–1400 ppb for Zn^2+^. The calibration plots for these three ions demonstrate good linear relationships between the response currents and their respective concentrations (Appendix A). The LOD for each metal ion is determined to be 10 ppb for Zn^2+^, 0.5 ppb for Cd^2+^, and 0.1 ppb for Pb^2+^ (S/N = 3, Appendix A). Compared to that of individual metal ions, these results reveal narrower linear ranges, lower sensitivities, and even relatively higher LOD, particularly for Cd^2+^ and Pb^2+^. This can potentially be attributed to the interactions among different metals, such as competitive deposition and intermetallic compound formation [43]. Nonetheless, this work concludes that the developed electrochemical platform holds great promise for practical applications in monitoring HMIs concentration in the environment. Significantly, the analytical performances of the resulting Bi-NPs@NC/GP in terms of LOD and linear range are comparable with and even better than other nanomaterials-based electrochemical sensors summarized in Appendix A.

### 3.5. Selectivity, Reusability, and Stability of the Integrated Electrochemical Sensor

Environmental samples often contain non-target HMIs or organic molecules that can potentially interfere with the detection results [44]. To assess the anti-interference ability of the proposed electrochemical method in the simultaneous determination of multiple HMIs, various interfering ions including K^+^, Ag^+^, Ca^2+^, Na^+^, Mg^2+^, Cl^−^, SO_4_^2−^, Cu^2+^, dopamine (DA), uric acid (UA), and ascorbic acid (AA) were added to solutions containing Zn^2+^, Cd^2+^, and Pb^2+^. Additionally, the concentrations of the interfering metal ions are maintained at ten times higher than the concentrations of the target analytes during the experiments. Figure 7B demonstrates that the introduction of non-target ions do not interfere with the simultaneous determination of Zn^2+^, Cd^2+^, and Pb^2+^. The results indicate that the developed electrochemical method exhibits excellent selectivity for the simultaneous analysis of these HMIs in the presence of high concentrations of various foreign ions. The amperometric current responses to 400 ppb metal ions decay approximately 4.4% for Pb^2+^, 8.1% for Cd^2+^, and 10.83% for Zn^2+^ of their initial values after storage at room temperature for 30 days (Figure 7C). Moreover, the response current deviations of seven electrodes prepared under the same conditions for testing 400 ppb Zn^2+^, Cd^2+^, and Pb^2+^ are within 1% (Figure 7D), indicating that the electrochemical sensor based on Bi-NPs@NC/GP exhibits good long-term stability and reproducibility.

### 3.6. Analysis of Real Samples in Tap and Lake Water

The applicability of the proposed electrochemical method has been evaluated by detecting the concentrations of Zn^2+^, Cd^2+^, and Pb^2+^ in real samples, such as tap water, Wuhan East Lake, and Yujia Lake, using Bi-NPs@NC/GP based sensor. No response of any target HMIs is detected in these samples, indicating that the concentrations of target analytes are very low or they are not present in the real samples. Subsequently, standard solutions of Zn^2+^, Cd^2+^, and Pb^2+^ are introduced to the real sample solutions to evaluate their recovery. The summarized results are presented in Table 2. The proposed electrochemical method is capable of simultaneously detecting Zn^2+^, Cd^2+^, and Pb^2+^ in real samples. The recoveries are 94.9–100.7% for Zn^2+^, 105–105.8% for Cd^2+^, and 105.5–106.1% for Pb^2+^, demonstrating that the proposed electrochemical method is highly accurate and feasible for the simultaneous detection of these metal ions in real samples.

## 4. Conclusions

In summary, we developed a high-performance Bi-NPs@NC/GP electrode material using a facile and controllable method by Bi-NPs@NC assembly derived from MOF as the highly active nanocatalyst and freestanding GP as the conductive and flexible electrode substrate, which is integrated in a portable electrochemical sensing system for ultra-sensitive detection of Pb^2+^, Cd^2+^, and Zn^2+^ in different environmental water samples. Our results demonstrate that in association with the uniform loading of Bi-NPs on carbon nanomaterials, the large specific surface area, and heteroatom doping of NC layer, the electrocatalytic activity and stability of the as-obtained Bi-NPs@NC/GP can be significantly improved, which further promote the overall performances and reliability of the sensing system, and can be used for real-time on-site detection of Pb^2+^, Cd^2+^, and Zn^2+^ in different real environmental samples. We envision that a handheld electrochemical sensor based on Bi-NPs@NC/GP electrode provides new insights into the design and construction of high-performance multifunctional sensing systems, which will contribute to the development of next-generation smart devices for environmental monitoring.

## Figures and Tables

**Figure 1 nanomaterials-13-02069-f001:**
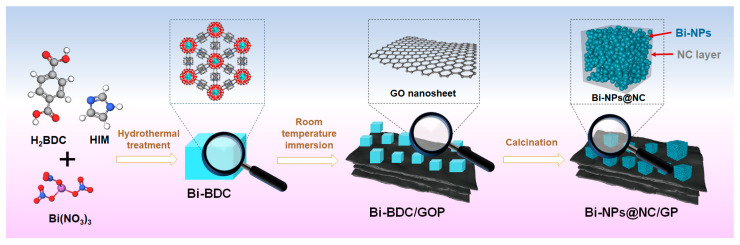
Schematic illustration for the preparation process of Bi-NPs@NC/GP.

**Figure 2 nanomaterials-13-02069-f002:**
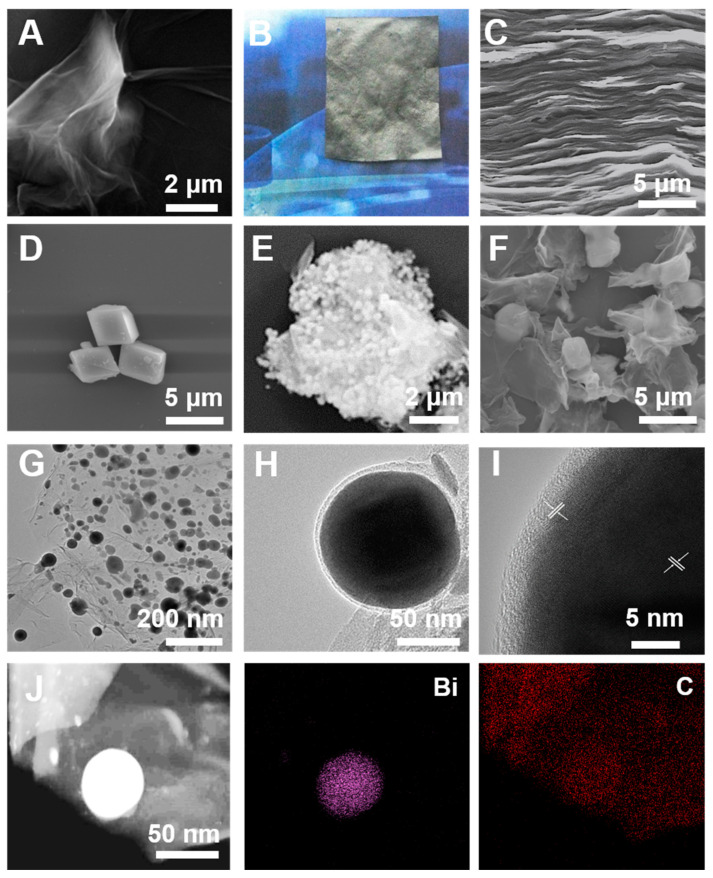
(**A**) SEM image of GO nanosheets; (**B**) photograph of GOP; (**C**) cross-section SEM image of GOP; (**D**) SEM image of Bi-BDC; (**E**) Bi-BDC/GOP; (**F**) Bi-NPs@NC assembly; (**G**) TEM; (**H**,**I**) HRTEM images; and (**J**) EDS elemental mapping images of Bi-NPs@NC/GP.

**Figure 3 nanomaterials-13-02069-f003:**
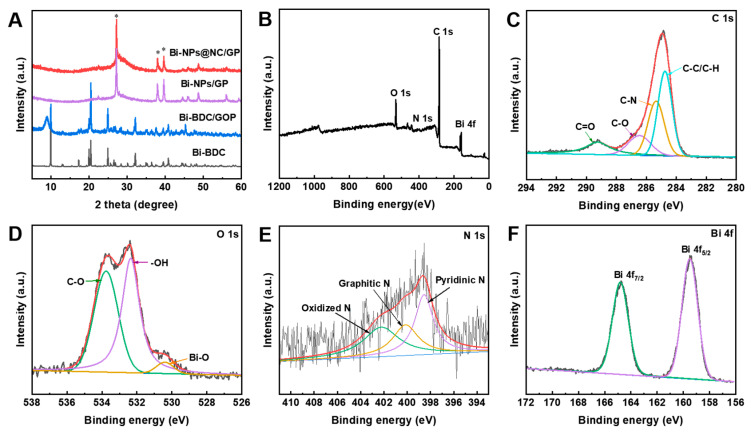
(**A**) XRD patterns of Bi-BDC, Bi-BDC/GOP, Bi-NPs/GP, and Bi-NPs@NC/GP; (**B**) XPS survey spectrum of Bi-NPs@NC/GP; XPS high-resolution (**C**) C 1s, (**D**) O 1s, (**E**) N 1s, and (**F**) Bi 4f spectrum of Bi-NPs@NC/GP.

**Figure 4 nanomaterials-13-02069-f004:**
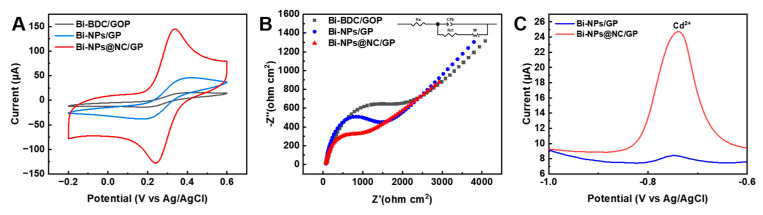
(**A**) CV curves of Bi-BDC/GOP, Bi-NPs/GP, and Bi-NPs@NC/GP in 0.15 M KCl solution containing 5 mM [Fe(CN)_6_]^3−/4−^, scan rate: 50 mV s^−1^; (**B**) EIS spectra of Bi-BDC/GOP, Bi-NPs/GP, and Bi-NPs@NC/GP in 0.15 M KCl solution containing 5 mM [Fe(CN)_6_]^3−/4−^, applied potential: 0.3 V vs. Ag/AgCl, frequency range: 0.01–10^6^ Hz; (**C**) SWASV curves of Bi-NPs/GP and Bi-NPs@NC/GP in 0.1 M ABS containing 400 ppb Cd^2+^.

**Figure 5 nanomaterials-13-02069-f005:**
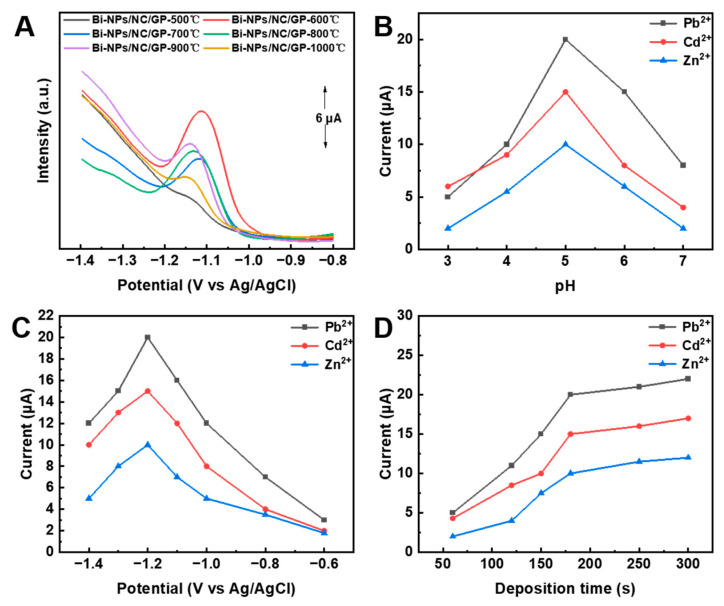
(**A**) SWASV curves of different electrodes in ABS containing 400 ppb Zn^2+^. Effect of (**B**) pH of the buffer electrolyte, (**C**) deposition potential, and (**D**) deposition time on SWASV response of 400 ppb Pb^2+^, 400 ppb Cd^2+^, and 400 ppb Zn^2+^ in ABS at Bi-NPs@NC/GP.

**Figure 6 nanomaterials-13-02069-f006:**
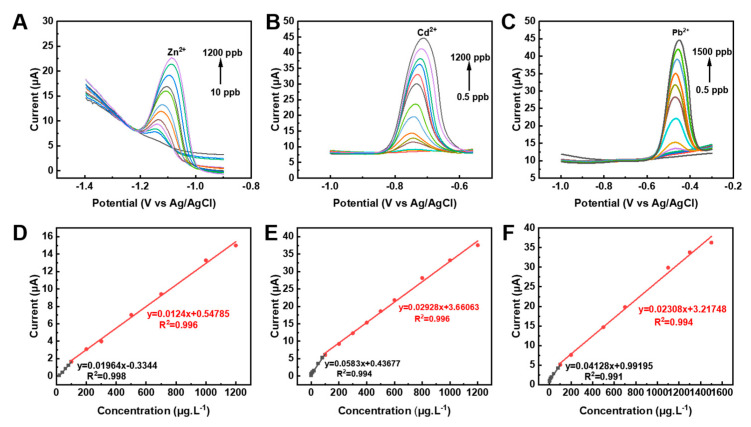
SWASV response of the Bi-NPs@NC/GP for the individual analysis of (**A**) Zn^2+^, (**B**) Cd^2+^, and (**C**) Pb^2+^ with varying concentrations in a range of 10–1200 ppb, 0.5–1200 ppb, and 0.5–1500 ppb, respectively. The linear calibration curves of oxidation current versus concentrations of (**D**) Zn^2+^, (**E**) Cd^2+^, and (**F**) Pb^2+^. Data are presented as mean ± SD from three independent measurements.

**Figure 7 nanomaterials-13-02069-f007:**
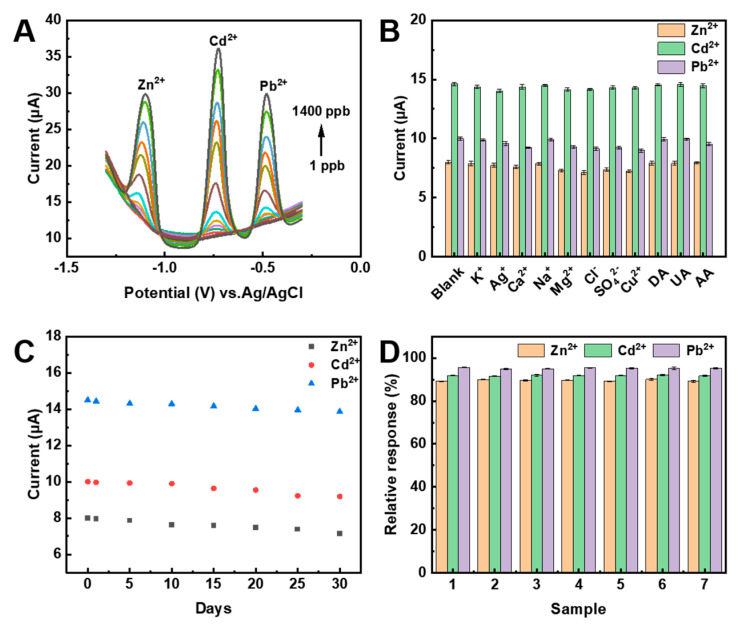
(**A**) SWASV response of the Bi-NPs@NC/GP for simultaneous analysis of Zn^2+^, Cd^2+^, and Pb^2+^ with varying concentrations in a range of 1–1400 ppb; (**B**) effect of interfering species on the determination of 400 ppb Zn^2+^, Cd^2+^, and Pb^2+^. The concentration of K^+^, Ag^+^, Ca^2+^, Na^+^, Mg^2+^, Cl^−^, SO_4_^2−^, Cu^2+^, DA, UA, and AA is 4 mg L^−1^; (**C**) relative current response of the sensor stored over 30 days toward 400 ppb Zn^2+^, Cd^2+^, and Pb^2+^; (**D**) relative current response of seven sensors prepared under the same conditions toward 400 ppb Zn^2+^, Cd^2+^, and Pb^2+^.

**Table 1 nanomaterials-13-02069-t001:** The analytical performances of Bi-NPs@NC/GP to detect individual Zn^2+^, Cd^2+^, and Pb^2+^ using SWASV under optimal conditions.

Separate Test	LOD (ppb)	Sensitivity (nA ppb^−1^)	Linear Range (ppb)
Zn^2+^	5	19.64	10–100
		12.4	100–1200
Cd^2+^	0.05	58.3	0.5–100
		29.28	100–1200
Pb^2+^	0.02	41.28	0.5–100
		23.08	100–1500
		0.689	1500–2.1 × 10^4^
		3.83	2.1 × 10^4^–3.1 × 10^5^
		1.81	3.1 × 10^5^–2.1 × 10^6^

**Table 2 nanomaterials-13-02069-t002:** Results of the detection of Zn^2+^, Cd^2+^, and Pb^2+^ in different environmental water samples by the proposed handheld electrochemical sensor based on Bi-NPs@NC/GP.

Sample	Metal Ions	The Proposed Method (ppb)			ICP-MS
		Spiked	Found	Recovery (%)	
Tap water	Zn^2+^	0	550.2	/	546
		50	600	100.7	
	Cd^2+^	0	1.6	/	1.2
		50	50.84	105	
	Pb^2+^	0	2.5	/	2
		50	52.11	105.5	
Wuhan East Lake	Zn^2+^	0	525	/	510
		50	550	98	
	Cd^2+^	0	5	/	3.8
		50	54	105.3	
	Pb^2+^	0	45	/	36
		50	88.2	106.1	
Yujia Lake	Zn^2+^	0	574	/	562.5
		50	584	94.9	
	Cd^2+^	0	10.25	/	9.64
		50	60.2	105.8	
	Pb^2+^	0	15	/	13.78
		50	54	105.8	

## Data Availability

The data presented in this study are available on request from the corresponding author.

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
