# Peer review of "Bi-MOF-Derived Carbon Wrapped Bi Nanoparticles Assembly on Flexible Graphene Paper Electrode for Electrochemical Sensing of Multiple Heavy Metal Ions"

_nanomaterials, 2023, doi:10.3390/nano13142069_

Round 1

Reviewer 1 Report

Comments on the manuscript “Bi-MOF derived carbon wrapped Bi nanoparticles assembly on flexible graphene paper electrode for electrochemical sensing of multiple heavy metal ions”.

                This manuscript claims that Bi-NPs@NC/GP electrode performance for multifunctional sensing systems for environmental monitoring. The manuscript contains electrochemical results using this electrode. The manuscript is not easy to understand by the way it has been presented. The following are specific comments.

1.       Page 3. The statement “Then GO paper (GOP) was prepared by printing method according to our previous work [22]. “Reference given is wrong. It should be 23. Also notice the sequence of numbering reversed in the references section.

2.       Figure 4(C) SWASV curves of Bi-NPs/GP containing 200 ppb Cd 2+. Note the peak potential is located at about -0.82 V. Figure 6B gives the peak potential at about -0.75 V. Why the peak potential values are different?

3.       Figure 5 (A) shows the peak potential for Cd2+ at about -1.1 V. This difference causes confusion on the claims made in the manuscript. What is the reason for such a huge shift?  

See the reference Journal of Hazardous Materials, Volume 456, 15 August 2023, Pages 131638. In this paper Cd2+ shows a peak at -0.82 V and Pb2+ at about -0.52 V. The reference electrode used is also the same (Ag/AgCl).

4.       Table 1 data on sensitivity does not agree with the data in Figures. See for example for Cd2+

in Figure 4, the current flow is about 24 µA for Cd2+ ion concentration of 200 ppb.

5.       Figure 6 D, E and F show positive intercepts. What is the cause for this intercept. If the Faradaic current is measured after subtracting the background current, the plot of current vs concentration should give zero intercept.

6.       Table 1 data is not supported by the data in Figure 6.

7.       Figure 7 and Figure 5 are contradictory. Why there is only one peak in Figure 5 and three peaks in Figure 7. If Figure 7 is representing individual runs for Cd2+, Zn2+ and Pb2+ it should be stated in the caption.

8.       Table 2 contains confusing data. It contains ICP-MS data with no literature references. If it is the data collected by the authors, there is no description of it in the manuscript.

This manuscript is not in an acceptable state.

Not easy to understand.

Author Response

Response to Reviewer 1 Comments

Point 1: Page 3. The statement “Then GO paper (GOP) was prepared by printing method according to our previous work [22].” Reference given is wrong. It should be 23. Also notice the sequence of numbering reversed in the references section.

Response 1: We greatly appreciate the reviewer for pointing out this mistake. Now the reference number in the text have been corrected in the revised manuscript.

Point 2Figure 4(C) SWASV curves of Bi-NPs/GP containing 200 ppb Cd 2+. Note the peak potential is located at about -0.82 V. Figure 6B gives the peak potential at about -0.75 V. Why the peak potential values are different?

Response 2: We greatly appreciate the reviewer for so careful reviewing of this manuscript. In fact, the peak potential of Cd2+ is located at about -0.75 V. With apologies for the mistake made in Figure 4C due to careless, we have now corrected Figure 4C in the revised manuscript. 

Figure 4. (A) CV curves of Bi-BDC/GOP, Bi-NPs/GP and Bi-NPs@NC/GP in 0.15 M KCl solution containing 5 mM [Fe(CN)6]3–/4–; Scan rate: 50 mV s−1. (B) EIS spectra of Bi-BDC/GOP, Bi-NPs/GP, and Bi-NPs@NC/GP in 0.15 M KCl solution containing 5 mM [Fe(CN)6]3–/4–; Applied potential: 0.3 V vs. Ag/AgCl; Frequency range: 0.01‒106 Hz. (C) SWASV curves of Bi-NPs/GP and Bi-NPs@NC/GP in 0.1 M ABS containing 400 ppb Cd 2+.

Point 3: Figure 5 (A) shows the peak potential for Cd2+ at about -1.1 V. This difference causes confusion on the claims made in the manuscript. What is the reason for such a huge shift?

Response 3: We have carefully checked this point and make a revision. We are quite sorry for that the graphs are incorrectly labeled. The oxidation peak located at -1.1 V in the square wave voltammetry plot of Figure 5A is attributed to Zn2+ ions, but not Cd2+ ions. Now this has been corrected in the revised manuscript. We have tried our best to check the whole manuscript and corrected all the errors. We sincerely hope the correction will meet with approval.

Figure 5. (A) SWASV curves of different electrodes in ABS containing 400 ppb Zn2+. Effect of (B) pH of the buffer electrolyte, (C) deposition potential and (D) deposition time on SWASV response of 400 ppb Pb2+, 400 ppb Cd2+, and 400 ppb Zn2+ in ABS at Bi-NPs@NC/GP.

Point 4: Table 1 data on sensitivity does not agree with the data in Figures. See for example for Cd2+ in Figure 4, the current flow is about 24 µA for Cd2+ ion concentration of 200 ppb.

Response 4: We sincerely appreciate the reviewer for the insightful comment. After retesting and analogy with Table 1, the above mistakes have been corrected with the following results:

Figure 4. (A) CV curves of Bi-BDC/GOP, Bi-NPs/GP and Bi-NPs@NC/GP in 0.15 M KCl solution containing 5 mM [Fe(CN)6]3–/4–; Scan rate: 50 mV s−1. (B) EIS spectra of Bi-BDC/GOP, Bi-NPs/GP, and Bi-NPs@NC/GP in 0.15 M KCl solution containing 5 mM [Fe(CN)6]3–/4–; Applied potential: 0.3 V vs. Ag/AgCl; Frequency range: 0.01‒106 Hz. (C) SWASV curves of Bi-NPs/GP and Bi-NPs@NC/GP in 0.1 M ABS containing 400 ppb Cd 2+.

Table 1. The analytical performances of Bi-NPs@NC/GP to detect individual Zn2+, Cd2+ and Pb2+ using SWASV under the optimal conditions.

Separate test

LOD (ppb)

Sensitivity

(nA ppb-1)

Linear range (ppb)

Zn2+

5

19.64

10-100

12.4

100-1200

Cd2+

0.05

58.3

0.5-100

29.28

100-1200

Pb2+

0.02

41.28

0.5-100

23.08

100-1500

0.689

1500-2.1×104

3.83

2.1×104-3.1×105

1.81

3.1×105-2.1×106

Point 5: Figure 6 D, E and F show positive intercepts. What is the cause for this intercept. If the Faradaic current is measured after subtracting the background current, the plot of current vs concentration should give zero intercept.

Response 5: We greatly appreciate the reviewer for the constructive comment. We think it is due to the background current of the electrode material. For SWV curves, the peak current is determined by taking the zero line as the baseline and the maximum current position as the peak, which is the conventional practice for peak finding. However, since we need to electrochemically deposit the analytes before each SWV test, which leads to poor agreement between the baseline and zero line, we perform the peak search by tangenting the baseline.

Point 6: Table 1 data is not supported by the data in Figure 6.

Response 6: We appreciate the reviewer for pointing out this problem. We have carefully checked Table 1 and Figure 6, and made the correction in then in the revised manuscript, as shown in the following:

Figure 6. SWASV response of the Bi-NPs@NC/GP for the individual analysis of (A) Zn2+, (B) Cd2+, and (C) Pb2+ with varying concentrations in a range of 10~1200 ppb, 0.5-1200 ppb, and 0.5-1500 ppb, respectively; The linear calibration curves of oxidation current versus concentrations of (D) Zn2+, (E) Cd2+, and (F) Pb2+. Data are presented as mean ± SD from three independent measurements.

Table 1. The analytical performances of Bi-NPs@NC/GP to detect individual Zn2+, Cd2+ and Pb2+ using SWASV under the optimal conditions.

Separate test

LOD (ppb)

Sensitivity

(nA ppb-1)

Linear range (ppb)

Zn2+

5

19.64

10-100

12.4

100-1200

58.3

0.5-100

Cd2+

0.05

29.28

100-1200

41.28

0.5-100

23.08

100-1500

Pb2+

0.02

0.689

1500-2.1×104

3.83

2.1×104-3.1×105

1.81

3.1×105-2.1×106

Point 7: Figure 7 and Figure 5 are contradictory. Why there is only one peak in Figure 5 and three peaks in Figure 7. If Figure 7 is representing individual runs for Cd2+, Zn2+ and Pb2+ it should be stated in the caption.

Response 7:  We are so sorry for that the caption of Figure 5 is incorrect. It should be “SWASV curves of different electrodes in ABS containing 400 ppb Zn2+”, but not “SWASV curves of different electrodes in ABS containing 200 ppb Pb2+, 200 ppb Cd2+, and 200 ppb Zn2+”. This has been corrected in page 8 of the revised manuscript.

Point 8: Table 2 contains confusing data. It contains ICP-MS data with no literature references. If it is the data collected by the authors, there is no description of it in the manuscript.

Response 8: We understand the concern of the reviewer. In this work, we have described the acquisition of ICP-MS data in section 2. Material and Methods. “The ICP-MS data were collected by preparing a certain concentration of ion standard solution, and calibrating it by ICP-MS to obtain a standard curve. Then we filtered and diluted the practical samples for ICP-MS measurement, and obtained their concentration values from intensities.” This has been added in page 4 of the revised manuscript.

We appreciate for Reviewers’ warm work earnestly and hope that the correction will meet with approval.

Thank you very much for your consideration in advance.

Sincerely yours,

Prof. Dr. Fei Xiao

School of Chemistry and Chemical Engineering

Huazhong University of Science & Technology

No. 1037 Luoyu Road, Wuhan, Hubei

Email: xiaofei@hust.edu.cn

Reviewer 2 Report

The present paper deals with the fabrication of sensors for electrochemical detection of heavy metal ions. The topic is important, the results are interesting and the structure of the sensor is indeed of nanostructural character, for which the manuscript suits weel for this journal.

The overall assessment of the manuscript is that the promising results are described in a heavily inadequate manner. Although the sensor fabrication is a bit heuristic, it causes less problem than the mode of the description of the experiments.

A, Major scientific issues (approx. in the order of importance)

1, One cannot see why the coating of the Bi nanoparticles becomes N-doped. The primary coating agent is a dicarboxilic acid without nitrogen, the nitrate ion (as the anion of the precursor) is removed and so is the rinsing agent, and the nitrogen gas in the annealing process is claimed to be inert to prevent oxidation. Therefore, the origin of N is to be unambiguously clarified. Is anything known about the role of the nitrogen? Was any attempt to prepare an N-free counterpart? Lines 220-223 tell that N contributes significantly to the electrode sensitivity. Such a statement can be made only in the case when a clear comparison basis is available.

2. A related issue is that the N signal in the XPS (Fig. 3) is very noisy, indicating a quite low signal-to-noise ratio. An estimation of the doping level would be required. The sub-peak identifications as "graphitic" and "pyridinic" are not fully clear since graphite is not a nitrogen-containing material.

3, The description of the electrochemical parameters is unclear. The peak separation (Lines 247-254) is irrelevant for electrical conductivity: the high-frequency intercept of the EIS curves at the real axis are quite the same (Fig. 4.b), hence, there is no hindrance in the electron transport.

4, The manuscript speaks about the oxidation of Cd2+ (Line 259). This is totally wrong.

5, Figure 4: a, the sweep rate is to be given. The difference in the base line currents as a function of the sweep rate is indicative of the capacitance effect, not the difference in electrical conductivity. b, The potential where the EIS curves were recorded must be given, also in relation of the onset of the equilibrium potential of the redox system used for the test.  Which form of the ferrocyanide ion was the precursor material? Perhaps both?

6, The initial section of the curves in Figs. 6 E and F are not linear. Especially in Fig. 6.F, the initial slope is about twice of the later one. Hence, the linear fit is by far not the best tool to describe the electrode response, and, additionally, it strongly distort the results in the entire concentration range studied.

B, Additional minor issues to be corrected

1, The abstract speaks about the "low detection limit, wide linear range and high sensitivity". How would you tell sensitivity from the low detection limit? It is suspicious that they are the same, instead, selectivity may be mentioned.

2, Lines 41-42: The text mentions a "complex form" of heavy metal ions. Complex would mean that a complexing agent is present, while the authors probably mean a multicomponent matrix (which is "complex" indeed in another sense, but this is the matrix, irrespectively of whether the metal ions themselves are complexed or not). Please clarify this problem.

3, It is the reviewer's opinion that the advantage of the electrochemical detection is overly emphasized in the Introduction (Lines 50-55). Even if the method in the present study is electrochemical, the description should be realistic.

4, Lines 43-44: Alloy formation is a problem only in the case when a solid deposit forms; however, in this case, metal atoms make the alloy, not metal ions. By the way, this can be the problem of the present system, too, since the deposition of either pure metals or alloys are equally possible but not studied in detail. Adsorption, in contrast, takes place in either ionic or metallic form, and it can be potentially unrelated to alloy formation.

5, Lines 286-284: Do the solubility products of the metal hydroxydes verify that a hydrolysis may take place? Is there any difference in the reduction rate if Me(OH)+ species are present instead of Me2+?

The check of the text has to be very careful, nearly more that a usual major revision, and the authors are strongly advised to take advantage of the help of a expert to make the manuscipt conformal to the publiction standards. Even though a large number of examples are marked in the annotated manuscript where corrections are needed, one cannot expect that the reviewer writes the paper instead of the authors themselves. Hence, the list of examples is to be taken strictly as representative but by far not complete.

One of the major problem is the use of present perfect where past tense is not only sufficient but required. The style unduly alternates between the two modes, which is very embarrassing and makes the reading very laborious.

Many problms are marked in the attached file (annotated manuscript).

Author Response

Response to Reviewer 2 Comments

  1. Major scientific issues (approx. in the order of importance)

Point 1: One cannot see why the coating of the Bi nanoparticles becomes N-doped. The primary coating agent is a dicarboxilic acid without nitrogen, the nitrate ion (as the anion of the precursor) is removed and so is the rinsing agent, and the nitrogen gas in the annealing process is claimed to be inert to prevent oxidation. Therefore, the origin of N is to be unambiguously clarified. Is anything known about the role of the nitrogen? Was any attempt to prepare an N-free counterpart? Lines 220-223 tell that N contributes significantly to the electrode sensitivity. Such a statement can be made only in the case when a clear comparison basis is available.

Response 1: According to the insightful comment of the reviewer, we have added more results and in-depth discussions in the revised manuscript. In this paper, Bi-MOF was prepared by adding imidazole, and the N of imidazole would coordinate with Bi during the self-assembly process of Bi-MOF. After being calcined at high temperature, Bi-MOF (i.e., Bi-BDC) was transferred into Bi-NPs wrapped by N doped carbon. Nevertheless, we can not synthesize N-free counterpart by the proposed method for comparison. Even so, the increase of the sensitivity of the electrode materials by introduction of N has been reported by previous reported work (J. Hazard. Mater. 2022, 442, 130020), which revealed that the heteroatoms doped carbon can lead to a strong aggregation of charges and will give a stronger adsorption towards analytes, thus enhancing the catalytic activity of Bi-NPs in Bi-NPs@NC/GP for specific catalytic reactions. This has been added in reference 24 and page 2 of the revised manuscript.

Point 2: A related issue is that the N signal in the XPS (Fig. 3) is very noisy, indicating a quite low signal-to-noise ratio. An estimation of the doping level would be required. The sub-peak identifications as "graphitic" and "pyridinic" are not fully clear since graphite is not a nitrogen-containing material.

Response 2: We sincerely appreciate the reviewer for the earnest work on review of this manuscript. In this work, the as-synthesized Bi-NPs@NC/GP contains graphitic N and pyridine N, the pyridine N is a N atom attached between two C atoms on the edge of graphene with a lone pair of electrons that can be oxidized, and the graphitic N refers to the N atom attached to three C atoms in the graphite base. However, the N dopant with an atomic concentration of 1 %, indicating a quite low doping level in Bi-NPs@NC, which results in a low signal-to-noise ratio. In our future work, we will pay more effort to deeply understand the role of heteroatom and try to solve this problem. Now this has been added in Page 6 of the revised manuscript.

Point 3: The description of the electrochemical parameters is unclear. The peak separation (Lines 247-254) is irrelevant for electrical conductivity: the high-frequency intercept of the EIS curves at the real axis are quite the same (Fig. 4b), hence, there is no hindrance in the electron transport.

Response 3: The reviewer is right that peak-to-peak separation is irrelevant for electrical conductivity, it should be an important index to evaluate the degree of redox process reversibility of the electrode, so the description has been revised as follows: In addition, the Bi-NPs@NC/GP demonstrates a peak-to-peak potential separation (ΔEp) of 69 mV, whereas that of Bi-BDC/GOP and Bi-NPs/GP are 156 mV and 141 mV, suggesting that Bi-NPs@NC/GP would greatly accelerate the heterogeneous electron transfer between the electrode and the [Fe(CN)6]3-/4- redox species. Figure 4B shows the characteristic Nyquist plots of Bi-BDC/GOP, Bi-NPs/GP, and Bi-NPs@NC/GP, fitted with the Randle equivalence circuit inset, where Rs is the resistance of solution (the intersection of the curve at real part Z0 in the high frequencies range), Rct is the Faradic charge transfer resistance (the semicircle at high frequencies), Cdl is the double layer capacitance and W is Warburg constant (the slope of the curves at a low frequency) [37]. The Rct values are calculated to be 1315 Ω, 861.9 Ω, and 285.7 Ω for Bi-BDC/GOP, Bi-NPs/GP, and Bi-NPs@NC/GP, respectively, verifying that Bi-NPs@NC can effectively promote electron transfer.” This has been corrected in Page 7 of the revised manuscript.

Point 4: The manuscript speaks about  (Line 259). This is totally wrong.

Response 4: Yes, the reviewer is right about this. The statement “the oxidation of Cd2+” is totally wrong. Actually, in this work, the anodic stripping voltammetry of heavy metal ions refers to the voltammetry method in which the metal ions are partially reduced to metal and dissolved into the electrode or precipitated on the electrode surface to form alloy at a certain potential. Then the reverse voltage is applied to the electrode to oxidize the metal on the electrode and generate oxidation current, and the analysis is performed according to the current-voltage curve of the oxidation process. Therefore, we have corrected it as “Figure 4C shows a strong stripping peak appeared at -0.75 V on Bi-NPs@NC/GP for Cd2+,” This has been corrected in Page 7 of the revised manuscript.

Point 5: Figure 4: a, the sweep rate is to be given. The difference in the base line currents as a function of the sweep rate is indicative of the capacitance effect, not the difference in electrical conductivity. b, The potential where the EIS curves were recorded must be given, also in relation of the onset of the equilibrium potential of the redox system used for the test. Which form of the ferrocyanide ion was the precursor material? Perhaps both?

Response 5: According to the comment of the reviewer, we have added these conditions during electrochemical process and made a revision in Figure 4. “As demonstrated in Figure 4A and Figure S2, a pair of symmetrical redox peaks are appeared on the CV curves of Bi-NPs@NC/GP, Bi-NPs@GP and Bi-BDC/GOP, which result from the reversible single electron redox reaction of [Fe(CN)6]3-/4-. The electrochemically active surface areas (ECSAs) of different modified electrodes are calculated by the Randles─Sevcik equation, which are 0.028, 0.036 and 0.12 cm2 for Bi-BDC/GOP, Bi-NPs/GP and Bi-NPs@NC/GP, respectively. In addition, the Bi-NPs@NC/GP demonstrates a peak-to-peak potential separation (ΔEp) of 69 mV, whereas that of Bi-BDC/GOP and Bi-NPs/GP are 156 mV and 141 mV, suggesting that Bi-NPs@NC/GP would greatly accelerate the heterogeneous electron transfer between the electrode and the [Fe(CN)6]3-/4- redox species.” This has been revised in Page 6-7 of the revised manuscript. Furthermore, the potential where the EIS curves were recorded must be given: prior to recording the EIS curve, we obtained the open-circuit potential of these samples, then setting this value as initial potential for EIS measurements.

Figure 4. (A) CV curves of Bi-BDC/GOP, Bi-NPs/GP and Bi-NPs@NC/GP in 0.15 M KCl solution containing 5 mM [Fe(CN)6]3–/4–; Scan rate: 50 mV s−1. (B) EIS spectra of Bi-BDC/GOP, Bi-NPs/GP, and Bi-NPs@NC/GP in 0.15 M KCl solution containing 5 mM [Fe(CN)6]3–/4–; Applied potential: 0.3 V vs. Ag/AgCl; Frequency range: 0.01‒106 Hz. (C) SWASV curves of Bi-NPs/GP and Bi-NPs@NC/GP in 0.1 M ABS containing 400 ppb Cd 2+.

Point 6: The initial section of the curves in Figure 6 E and F are not linear. Especially in Fig. 6.F, the initial slope is about twice of the later one. Hence, the linear fit is by far not the best tool to describe the electrode response, and, additionally, it strongly distort the results in the entire concentration range studied.

Response 6: We agree with the reviewer. According to the reviewer's comment, we have divided the linear regression equations for Zn2+, Cd2+, and Pb2+ into two curves, one for low concentrations and another for high concentrations. For example, consider the linear regression equation for Zn2+ in the concentration range of 10-100 ppb: y = 0.01964x - 0.3344. In the concentration range of 100~1200 ppb, the equation becomes y = 0.0124x + 0.54785. This division is based on the understanding that at low concentrations, the electrode adsorption of ions follows a unimolecular layer adsorption process. However, at high concentrations, the adsorption process transitions into a multilayer adsorption mechanism. This distinction has been supported by previous research, and it justifies the need for two different linear equations to accurately represent the relationship between current and concentration for these analytes.

Figure 6. SWASV response of the Bi-NPs@NC/GP for the individual analysis of (A) Zn2+, (B) Cd2+, and (C) Pb2+ with varying concentrations in a range of 10-1200 ppb, 0.5-1200 ppb, and 0.5-1500 ppb, respectively; The linear calibration curves of oxidation current versus concentrations of (D) Zn2+, (E) Cd2+, and (F) Pb2+. Data are presented as mean ± SD from three independent measurements.

B, Additional minor issues to be corrected

Point 1: The abstract speaks about the "low detection limit, wide linear range and high sensitivity". How would you tell sensitivity from the low detection limit? It is suspicious that they are the same, instead, selectivity may be mentioned.

Response 1: We greatly appreciate the reviewer for so careful reviewing of our article. Sensitivity means the degree of change in the amount of response due to a change in unit concentration of the substance to be measured. It can be described by the ratio of the instrument's response or other indicative amount to the corresponding concentration. The limit of detection (LOD) is the lowest detectable concentration that can be distinguished from noise when the sample is extracted, processed, and detected according to the requirements of the analytical method. The ideally LOD corresponds to a response value that is at least 3~5 times higher than the noise. Figure 7B demonstrates that the introduction of non-target ions do not interfere with the simultaneous determination of Zn2+, Cd2+, and Pb2+, indicative of the developed electrochemical method exhibits excellent selectivity for the simultaneous analysis of these HMIs in the presence of high concentrations of various foreign ions. On this basis, we made a revision as follows: “The result shows that Zn2+, Cd2+, and Pb2+ can be detected with extremely low detection limits, wide linear range, high sensitivity as well as good selectivity.” This has been corrected in Page 1 of the revised manuscript.

Point 2: Lines 41-42: The text mentions a "complex form" of heavy metal ions. Complex would mean that a complexing agent is present, while the authors probably mean a multicomponent matrix (which is "complex" indeed in another sense, but this is the matrix, irrespectively of whether the metal ions themselves are complexed or not). Please clarify this problem.

Response 2: We sincerely appreciate the reviewer for the insightful comment. “It is well known that HMIs in natural water bodies typically appear in a complex form [9,10], i.e., multiple heavy metal ions coexist to form intermetallic compounds that compete for adsorption at the active site, which leads to a more limited simultaneous detection.” This has been corrected in Page 1 of the revised manuscript.

Point 3: It is the reviewer's opinion that the advantage of the electrochemical detection is overly emphasized in the Introduction (Lines 50-55). Even if the method in the present study is electrochemical, the description should be realistic.

Response 3: We sincerely appreciate the reviewer for the earnest work on review of this manuscript, and have checked the reviewer’s comment carefully. “Due to the inherent advantages of low cost, fast response, easy-to-use and miniaturization, the electrochemical sensors can fit well with simultaneous analysis of multiple HMIs [15,16].” This has been corrected in Page 2 of the revised manuscript. 

Point 4: Lines 43-44: Alloy formation is a problem only in the case when a solid deposit forms; however, in this case, metal atoms make the alloy, not metal ions. By the way, this can be the problem of the present system, too, since the deposition of either pure metals or alloys are equally possible but not studied in detail. Adsorption, in contrast, takes place in either ionic or metallic form, and it can be potentially unrelated to alloy formation.

Response 4: We greatly appreciate the reviewer for so careful reviewing of our article. Alloy formation is a problem only in the case when a solid deposit forms; however, in this case, metal atoms make the alloy, not metal ions. In our work, owing to the synergistic effect between the high conductivity and large surface area of GP, numerous accessibility of Bi-NPs, and abundant active sites of NC, Bi-NPs@NC/GP behave adsorption towards HMIs (the first step of HMI detection), followed by partial reduction to form metal atoms that either dissolve into the microelectrode or precipitate on the electrode surface at a specific potential to form alloy (the second step of HMI detection), then render redox kinetics more favorable. Thus, adsorption, takes place in ionic form or metallic form, and it is the first step in the anodic stripping voltammetry of heavy metal ions. This has been corrected in Page 1 of the revised manuscript. 

Point 5: Lines 286-284: Do the solubility products of the metal hydroxydes verify that a hydrolysis may take place? Is there any difference in the reduction rate if Me(OH)+ species are present instead of Me2+?

Response 5: We understand the concern of the reviewer. The dissolution products of metal hydroxides have been verified to undergo hydrolysis by some literatures, and they are introduced in the paper to further prove this. “In addition, pH value ranging from 3.0 to 7.0 has been evaluated (Figure 5B), and the results suggest that the optimal pH of the buffer solution is 5.0. The possible reason for the decrease in peak currents at relatively lower pH values (i.e., pH 3.0 to 5.0) is due to the protonation of hydrophilic groups, which weaken the absorption of HMIs to the electrode materials. At relatively high pH values (i.e., pH 5.0 to 7.0), the decrease in current signal is attributed to the hydrolysis of HMIs, which hinder the accumulation of HMIs at the electrode surface (J. Phys. Chem. C 2012, 116, 1, 1034–1041, Anal. Chem. 2018, 91, 1, 888-895”. This has been corrected in Page 7 and added as reference 40 and 41 of the revised manuscript.

We appreciate for Reviewers’ warm work earnestly and hope that the correction will meet with approval.

Thank you very much for your consideration in advance.

Sincerely yours,

Prof. Dr. Fei Xiao

School of Chemistry and Chemical Engineering

Huazhong University of Science & Technology

No. 1037 Luoyu Road, Wuhan, Hubei

Email: xiaofei@hust.edu.cn

Round 2

Reviewer 2 Report

The authors considered the suggestions made on the original version, as a result of which the quality of the manuscript is much better now.

A few comments remained that are partly substantial and partly formal. All these comments can be seen in the attached file. The more important (and indispensable) modification required is that at least a part of the explanation on the origin of the N content of the samples should be also added to the manuscript text. It was only the part of the authors' reply, but the main point is that the reader must also be allowed to see it.

All comments are included into the file attached.

Author Response

We greatly appreciate the reviewer for so careful reviewing of this manuscript. Accordingly, we have checked the whole paper and corrected all the mistakes, and also added more results in the revised manuscript. 

Furthermore, we agree with the reviewer that more important (and indispensable) modification required is that at least a part of the explanation on the origin of the N content of the samples should be also added to the manuscript text. Therefore, we have added more in-depth discussions:”Moreover, in this work, Bi-MOF was prepared by adding imidazole, and the N of imidazole would coordinate with Bi during the self-assembly process of Bi-MOF. After being calcined at high temperature, Bi-MOF (i.e., Bi-BDC) was transferred into Bi-NPs wrapped by N doped carbon.” and  The atomic concentration of N in Bi-NPs@NC/GP is1%“. This has been added in page 6 of the revised manuscript. 

Some other minor issues marked by reviewers have been corrected in the manuscript.

We appreciate for Editors and Reviewers’ warm work earnestly and hope that the correction will meet with approval.

Thank you very much for your consideration in advance.